# The Role of Tocotrienol in Arthritis Management—A Scoping Review of Literature

**DOI:** 10.3390/ph16030385

**Published:** 2023-03-02

**Authors:** Hashwin Singh Tejpal Singh, Alya Aqilah Aminuddin, Kok-Lun Pang, Sophia Ogechi Ekeuku, Kok-Yong Chin

**Affiliations:** 1Department of Pharmacology, Faculty of Medicine, Universiti Kebangsaan Malaysia, Kuala Lumpur 56000, Malaysia; 2Newcastle University Medicine Malaysia, Iskandar Puteri 79200, Malaysia

**Keywords:** cartilage, chondrocytes, joint, osteoarthritis, rheumatoid arthritis, vitamin E

## Abstract

Arthritis is a cluster of diseases impacting joint health and causing immobility and morbidity in the elderly. Among the various forms of arthritis, osteoarthritis (OA) and rheumatoid arthritis (RA) are the most common. Currently, satisfying disease-modifying agents for arthritis are not available. Given the pro-inflammatory and oxidative stress components in the pathogenesis of arthritis, tocotrienol, a family of vitamin E with both anti-inflammatory and antioxidant properties, could be joint-protective agents. This scoping review aims to provide an overview of the effects of tocotrienol on arthritis derived from the existing scientific literature. A literature search using PubMed, Scopus and Web of Science databases was conducted to identify relevant studies. Only cell culture, animal and clinical studies with primary data that align with the objective of this review were considered. The literature search uncovered eight studies investigating the effects of tocotrienol on OA *(n* = 4) and RA (*n* = 4). Most of the studies were preclinical and revealed the positive effects of tocotrienol in preserving joint structure (cartilage and bone) in models of arthritis. In particular, tocotrienol activates the self-repair mechanism of chondrocytes exposed to assaults and attenuates osteoclastogenesis associated with RA. Tocotrienol also demonstrated strong anti-inflammatory effects in RA models. The single clinical trial available in the literature showcases that palm tocotrienol could improve joint function among patients with OA. In conclusion, tocotrienol could be a potential anti-arthritic agent pending more results from clinical studies.

## 1. Introduction

Arthritis is the inflammation or degeneration of one or more joints. It affects people of all ages, including children. It manifests commonly as joint pain and stiffness, but other specific symptoms are possible, depending on the type of arthritis. There are more than 100 types of arthritis, whereby osteoarthritis (OA) and rheumatoid arthritis (RA) are the two most common types [1]. Other types include gout, psoriatic arthritis, juvenile arthritis, degenerative arthritis and ankylosing spondylitis [2].

OA is a progressive degenerative joint disease resulting in pain and disability. It is mainly characterized by cartilage degeneration and joint space narrowing [3]. Aside from the cartilage, the pathological changes in OA also involve other joint structures, such as the meniscus, synovial membrane and infrapatellar fat pad [4]. The Global Burden of Disease (GBD) report indicated that the worldwide prevalence of OA was 528 million in 2019, representing a 27.5% increase from 2010 [5,6]. Similarly, the years of healthy life lost to disability for OA also showed a 27.5% increase from 2010 [5]. OA has been linked to other morbidities, such as Alzheimer’s disease [7] and fractures [8].

In OA, the viability of chondrocytes is compromised by biomechanical injury, inflammation and oxidative stress [9,10]. These stressors reprogram the chondrocytes to hypertrophic phenotype to compensate for the cartilage catabolic process transpiring in the joint [9,10]. Nuclear factor-κB (NF-κB) and mitogen-activated protein kinase (MAPK) play vital roles in chondrocyte reprogramming [11,12]. The recurrent inflammation and cartilage destruction form a self-propagating loop that stimulates the OA to progress further [13] (Figure 1A).

On the other hand, RA is an autoimmune disease involving the production of autoantibodies (rheumatoid factor and anti-citrullinated protein antibody (ACPA)); persistent inflammation and hyperplasia of the synovial membrane; destruction of cartilage and bone; and other manifestations, including the skeletal, cardiovascular and pulmonary systems [14]. RA is commonly linked to multi-system complications, progressive impairment, premature mortality and socioeconomic burden [15]. The GBD in 2019 indicated a worldwide prevalence of RA of 18.6 million, which was a 22.3% increase from 2010. Furthermore, the years lost due to a disability of RA also showed an upward trend with a 21.8% increase since 2010 [16].

The pathogenesis of RA involved the interactions of genetic and environmental factors and autoimmune-initiating agents at various anatomical locations, including the oral cavity, respiratory tract and digestive system [17]. The initiation of RA encompasses self-citrullination of proteins, such as citrullinated proteins, including Epstein–Barr nuclear antigen 1, fibrin and collagen type II, which can be found throughout the body, including the joints [17,18]. This event ultimately stimulates the production of autoantibodies, such as anti-citrullinated protein/peptide antibodies (ACPA), rheumatoid factor and anti-Epstein–Barr nuclear antigen-1 antibodies [19]. The ensuing inflammation is responsible for the systemic manifestation of the disease [20] (Figure 1B).

Currently, there is no effective disease-modifying agent available for OA. The standard treatments primarily serve to relieve the symptoms of OA, such as non-steroidal anti-inflammatory drugs (NSAIDs), paracetamol and intra-articular corticosteroid injections [21]. These treatment options also come with unfavorable adverse effects that affect the cardiovascular, gastrointestinal and renal systems [21,22]. The effectiveness of slow-acting disease-modifying osteoarthritic agents, such as glucosamine sulphate, chondroitin and avocado or soybean unsaponifiables, in treating OA, is still inconclusive [23]. Patients with end-stage OA often require arthroplasty [24], which is invasive, costly and comes with risks of complications such as infections [25,26].

Similarly, the management of RA is directed towards managing pain and inflammation; improving strength and mobility; and preventing joint destruction and deformity [27]. The first-line drugs for RA include NSAIDs and corticosteroids, but they possess severe side effects that affect gastrointestinal, pulmonary and hematological systems [27,28]. The second-line drugs for RA, disease-modifying antirheumatic drugs (DMARDs), aim to delay or halt joint damage [29]. The most widely prescribed DMARD is methotrexate, which is an immunosuppressant and hepatotoxic agent [30,31]. Biologics are emerging targeted medications to treat RA [32]. Nonetheless, they may increase susceptibility to infections and neurologic diseases, such as multiple sclerosis and lymphoma [33,34].

Given how oxidative damage and inflammation play key roles in the onset and progression of OA and RA, it can be hypothesized that bioactive compounds that possess antioxidant and/or anti-inflammatory properties could be useful in preventing joint degeneration or damage from chronic inflammatory processes in OA and RA. Tocotrienol, a subfamily of vitamin E, has demonstrated both anti-inflammatory and antioxidant effects [35]. Tocotrienol exists in four known isoforms, which are α-, β-, γ- and δ-tocotrienol. Tocotrienol can be found naturally in various compositions in oil derived from annatto seed, palm kernel, rice bran, sunflower seed, flax seed, poppy seed, grapefruit seed and others [35,36]. Studies have shown that γ-tocotrienol has been effective at preventing the hydrogen peroxide-induced decrease in the activity of antioxidative enzymes and apoptosis of osteoblasts [37]. Tocotrienol also suppresses pro-inflammatory cytokines in animal models of metabolic syndrome [38,39] and osteoporosis [40].

Tocotrienol could be a potential treatment agent for OA and RA, given its antioxidant and anti-inflammatory properties. However, previous discussion on the protective effects of vitamin E on joints largely focuses on α-tocopherol, the most common vitamin E isomer in nature. Given its superior biological activities over α-tocopherol [35,41], tocotrienol could be a better anti-arthritis agent. Therefore, the current review aims to summarize the effects of tocotrienol on joint health from the current literature. The research gaps and potential future research on this topic are also discussed. We hope this review will encourage the practical implementation of tocotrienol in the treatment and management of RA and OA.

## 2. Methodology

This scoping review was constructed according to the framework of Arksey and O’Malley [42] and in compliance with Preferred Reporting Items for Systematic Reviews and Meta-Analyses extension for scoping reviews (Appendix A) [43]. The following steps were adopted: (1) identifying the research question; (2) identifying the relevant studies; (3) study selection; (4) charting the data; (5) collating, summarizing and reporting the results.

### 2.1. Identifying the Research Question

The research question was: What are the effects of tocotrienol on joint health? Although the review focuses on tocotrienol, it is understood that natural tocotrienol mixtures commonly used in studies contain tocopherols, and they were considered [44]. Joint health is a broad term, which covers protection against arthritis or any joint structural deterioration, particularly at the cartilage. All studies reporting joint-related outcomes were considered.

### 2.2. Identifying Relevant Studies

A literature search was performed on electronic databases (PubMed, Scopus and Web of Science) in January 2023 using the search string: tocotrienol AND (arthritis OR joint OR cartilage OR chondrocytes). In this search, all primary studies involving in vitro and animal models, or humans, investigating the effects of tocotrienol on joint health, were considered. Articles without primary results, such as reviews, perspectives, commentary, letters to the editor, books and book chapters, were not considered. Conference abstracts and proceedings were not included due to incomplete data and redundancy. Articles not written in English were excluded. No additional filter was applied during the search.

### 2.3. Study Selection

The literature was organized using Endnote (version 20.4, Clarivate, London, UK) [45]. The search results were downloaded from the three electronic databases. Duplicated items were removed using Endnote and checked manually. K.Y.C. and S.O.E. screened the titles and abstracts for relevant studies. Next, the full texts of the selected articles were obtained and screened based on inclusion and exclusion criteria. The reference list of the included articles was screened to identify literature that was missed during the search. Any discrepancies were resolved by discussion, and opinions were sought from the other authors. The article selection process is summarized in Figure 2.

### 2.4. Charting the Data

H.S.T.S. and A.A.A. extracted relevant information from the selected studies, which included researchers, publication years, study design (subjects or disease models used, type of tocotrienol, dosage, treatment period) and major findings using a standard Excel table.

### 2.5. Collating, Summarising and Reporting the Results

Due to the heterogeneity of the studies involved and the variables of interest reported, the scoping review approach was adopted to report the search results, instead of synthesizing any particular variables. The main purpose of a scoping review is to provide an overview of the progress in the field. In this regard, the study types, disease models, tocotrienol (dose and treatment period) and major outcomes are summarized and reported. The role of tocotrienol in protecting joint health in arthritic individuals and the research gaps identified are discussed.

## 3. Results

The literature search using the three electronic databases found 107 items (PubMed = 23, Scopus = 38 and Web of Science = 46). After removing 33 duplicates, 74 items were subjected to title and abstract screening. Sixty-seven items were excluded based on various reasons (not related = 39, no primary data = 27 and conference item = 1). Seven items were subjected to full-text screening, and none was eliminated at this stage. One additional item was added while reviewing the reference list of the included articles. Overall, we found four studies on the effects of tocotrienol on OA and four studies on RA (Figure 1).

### 3.1. OA Study

Among the OA studies, one involved oral supplementation of palm tocotrienol (400 mg/kg/day for 6 months) among patients with OA (Kellgren–Lawrence score of 2 and 3) [46]; two studies involved oral supplementation of palm (100 mg/kg/day for 4 weeks) or annatto tocotrienol (50–150 mg/kg/day for 4 weeks) in rats with monosodium iodoacetate-induced OA [47,48]; another study used SW1353 chondrosarcoma cells exposed to monosodium iodoacetate and palm (25–50 µg/mL for 24 h) or annatto tocotrienol (10–20 µg/mL for 24 h) in the experiment [49]. Glucosamine sulphate was used as the positive control in the in vivo studies [46,47,48].

The in vitro study demonstrated that tocotrienol, especially the annatto derivative, could elicit the cellular protective mechanism of SW1353 chondrocytes against damage by monosodium iodoacetate. This was evidenced by an increased collagen II type α1 to collagen I type α1 ratio, and increased SRY-box transcription factor 9 and aggrecan expression in chondrocytes exposed to both monosodium iodoacetate and annatto tocotrienol [49]. These markers indicated the differentiation and functionality of chondrocytes [50,51]. Furthermore, both palm and annatto tocotrienol also ameliorated oxidative stress in chondrocytes exposed to monosodium iodoacetate. However, the same study demonstrated that pretreatment of chondrocytes with both forms of tocotrienol for 24 h before monosodium iodoacetate did not protect the cells from damage [49].

In the animal study, rats with OA treated with annatto tocotrienol (100–150 mg/kg body weight) showed attenuation of cartilage degradation and circulating cartilage oligomeric matrix protein (COMP) and hyaluronan levels. Annatto tocotrienol at 150 mg/kg/body weight also significantly reduced the osteoclast number in subchondral bone and circulating osteocalcin levels in rats, probably indicating reduced high bone remodeling in subchondral bone [48]. In another animal study, rats with monosodium iodoacetate-induced OA were treated with palm tocotrienol mixture, glucosamine sulphate or a combination of both for 4 weeks. The combination improved body weight and grip strength better than individual treatments. All treatment regimens also reduced cartilage degradation and circulating COMP levels in rats with OA [47].

The efficacy of palm tocotrienol (400 mg daily for 6 months) has been compared with that of glucosamine sulphate (1500 mg daily for 6 months) in patients with OA. Both palm tocotrienol and glucosamine sulphate reduced the visual analogue score for standing and walking, and the Western Ontario and McMaster Universities’ OA index score for the patients. On the other hand, palm tocotrienol reduced serum malondialdehyde better than the glucosamine sulphate group [46].

A summary of studies investigating the effects of tocotrienol on OA models is presented in Table 1.

### 3.2. RA Study

Among the studies on RA, three used rats receiving an intradermal injection of collagen-complete Freund’s adjuvant mixture in the tail or/and paws [52,53,54]. The rats were treated with γ-tocotrienol (5 mg/kg body weight), δ-tocotrienol (10 mg/kg body weight) and palm tocotrienol (30 mg/kg body weight) after joint inflammation occurred (day 21 until day 45 post-induction). Another study investigated the effects of tocotrienol in mediating the relationship among fibroblast-like synoviocytes (FLS) obtained from patients with RA undergoing total knee replacement surgery, T helper 17 cell (Th17) differentiation and osteoclastogenesis in blood peripheral mononuclear cells (PBMCs) in vitro [55].

In the cellular study, tocotrienol was reported to prevent Th17 formation from PBMCs and soluble receptor activator of nuclear factor kappa-Β ligand (RANKL) production. Tocotrienol suppressed Th17-mediated RANKL and tumor necrosis factor-α (TNF-α) expression in FLS [55]. Tocotrienol was suggested to mediate these effects via suppression of mammalian target of rapamycin (mTOR), extracellular signal-regulated kinase (ERK) and inhibitor of kappa B-α (IκBα) pathways while increasing interleukin-17 (IL-17) activated phosphorylation of adenosine monophosphate-activated protein kinase (AMPK) in FLS [55]. It also reduced RANKL/Th-17/IL17-treated, FLS-mediated PBMCs’ osteoclastogenesis, as indicated by reduced osteoclast differentiation markers, such as tartrate-resistant acid phosphatase (TRAP), cathepsin K, dendritic cell-specific transmembrane protein (DC-STAMP), nuclear factor of activated T cells 1 (NF-ATc1) and osteoclast stimulatory transmembrane protein (OC-STAMP) [55]. These findings indicated that tocotrienol could prevent bone resorption associated with RA. However, this study does not disclose which tocotrienol isomer was used.

The three animal studies reported that a γ-tocotrienol, δ-tocotrienol and palm tocotrienol mixture could prevent paw oedema induced by collagen-complete Freund’s adjuvant mixture [52,53,54]. Gamma-tocotrienol and δ-tocotrienol also reduced degenerative histological changes at the joint, such as joint space narrowing and inflammation [53,54]. Apart from protective effects on the cartilage, the palm tocotrienol mixture preserved bone health in rats with RA [52]. All three forms of tocotrienol lowered circulating inflammatory markers in rats with RA [52,53,54]. Gamma-tocotrienol also improved the redox status of the rats [54].

We did not find any clinical trial related to this topic. A summary of studies investigating the effects of tocotrienol on RA is presented in Table 2.

## 4. Discussion

### 4.1. Effects on OA

In early OA, activated chondrocytes, synoviocytes and mononuclear cells secrete IL-1β and TNF-α, which are known pro-inflammatory cytokines [56,57,58]. These cytokines then induce inflammation in joint tissues, which subsequently release IL-6, IL-1β and TNF-α. IL-1β has been known to cause the degeneration of articular cartilage by interfering with the synthesis of MMP-1 and MMP-13 by chondrocytes [59]. IL-1β is also a driver of oxidative damage in OA due to its role in stimulating the production of reactive oxygen species (ROS) [60]. IL-1β also initiates a cascade of reactions which result in the activation of the pro-inflammatory NF-κB signaling pathway. Moreover, IL-1β stimulates the expression of TNF-α and surface expression of TNF receptor in chondrocytes [61]. IL-1β drives aggrecan degradation, as it stimulates the production of an aggrecanase known as a disintegrin and metalloproteinase with thrombospondin motif (ADAMTS5). IL-17, an inflammatory cytokine that is secreted by Th17, mast cells and myeloid cells, is elevated in the serum of OA patients [62]. IL-17 results in cartilage degradation and favors inflammation by stimulating synovial fibroblasts and chondrocytes to release IL-6, IL-8 and TNF-α [63].

Osteoarthritic joint tissues also release IL-6, which ultimately binds to the soluble IL-6 receptor, causing trans-signaling [64]. This activates the immune system and results in the recruitment of mononuclear cells such as monocytes to the inflamed joint [64]. The increased expression of the macrophage colony-stimulating factor receptor induced by IL-6 trans-signaling also distorts monocyte-to-macrophage differentiation [65]. IL-8 also plays a role in OA, where it recruits and activates neutrophils to secrete elastase, which then destroy collagen 2 crosslinks and proteoglycan in the articular cartilage [66]. Human osteoarthritic cartilage explants cultured with a conditioned medium containing macrophages expressing interferon-γ and TNF-α expressed higher levels of pro-inflammatory cytokines, such as IL-1β, IL-6, MMP-13 and ADAMTS5; and lower levels of cartilage matrix elements, such as aggrecan and collagen 2 [67].

Tocotrienol has been extensively reported to reduce circulating inflammation markers in previous studies [38,68,69,70]. However, in the current literature, the effects of tocotrienol on inflammatory markers and immune cells in the joint space of animals and patients with OA have not been studied. This aspect is important, as inflammation in OA is localized, and systematic markers may not represent the degree of inflammatory status in the joint.

Neutrophils, which modulate inflammatory activities by producing various inflammatory factors in OA, are also activated and recruited to sites of inflammation by IL-17 [71]. Elevated levels of IL-17 and IL-22 have been observed in the temporomandibular joints of OA patients, which ultimately leads to an increase in RANKL [72]. This elevation in RANKL is responsible for the induction of osteoclast differentiation and the resorption of subchondral bone, which is a pathognomonic feature of OA [72]. The study by Chin et al. demonstrated that annatto tocotrienol could reduce osteoclast formation in the subchondral bone [48]. This effect could potentially prevent the osteolytic lesion commonly observed in early OA [73]. The anti-osteoclastogenic effects of tocotrienol have been demonstrated in vitro and in animal models of osteoporosis [44,74,75,76]. The effects could be extended to the prevention of subchondral bone osteolytic lesions in OA. However, in the study of Chin et al., structural preservation of subchondral bone was not demonstrated via histomorphometry or micro-computed tomography. Hence, the prevention of osteolytic lesions could not be confirmed.

Alongside inflammation, oxidative stress also plays an important role in the development and progression of OA [77]. Oxidative stress in chondrocytes with excessive production of ROS is one of the main drivers in the pathological changes seen in OA [77]. In OA, IL-1β upregulates ROS-producing pathways, including inducible nitric oxide synthase and cyclooxygenase-2. This results in an increased production of nitric oxide (NO) and prostaglandin E2 (PGE2) [78,79]. Chondrocyte proliferation and extracellular matrix synthesis are inhibited by PGE2, and collagen 2 and proteoglycan synthesis are inhibited by NO [79,80]. Furthermore, NO and PGE2, ADAMTS4 and ADAMTS5 were also elevated in OA patients, which further promoted ECM degeneration and inflammatory factor production [81,82].

The current evidence shows that tocotrienol has a protective effect on chondrocytes and/or joint tissue in OA. As evidenced in the literature, tocotrienol improved the redox status of the body, as evidenced by the lowering of serum malondialdehyde level, an effective biomarker of lipid peroxidation, after 6-month supplementation in patients with OA [46]. Tocotrienol’s antioxidant properties were further showcased when osteoarthritic chondrocytes exposed to monosodium iodoacetate that were treated with annatto and palm tocotrienol showed complete suppression of 8-isoprostane F2-α, a stable lipid peroxidation marker [49]. Previous studies also showed improvement in systemic and skeletal markers of oxidative stress following tocotrienol supplementation [40,83].

Additionally, annatto tocotrienol also induced an increase in collagen type II α1/collagen type I α1 in the chondrocyte cell line exposed to monosodium iodoacetate [49]. This indicates that annatto tocotrienol could promote the self-repair mechanism of chondrocytes and reduce the degenerative changes observed in OA, given that collagen type II acts as a marker for well-differentiated chondrocytes and collagen type I acts as a marker for de-differentiated chondrocytes [84]. The animal studies included in the current review also showed improvements in the cartilage structure and joint function in OA models [47,48]. The tocotrienol-treated rats had reduced COMP levels [47,48] and hyaluronic acid [48], products of cartilage breakdown when compared to unsupplemented rats, which signifies lower rates of cartilage degradation. The human clinical trials also illustrated similar improvement in joint function, but structural changes were not studied [46].

A summary of the effects of tocotrienol on OA is presented in Figure 3A.

### 4.2. Effects on RA

RA is an autoimmune disease with multiple bilateral joint involvements. Th17 cells are known as drivers of RA, particularly at the early stages of disease progression [85,86,87]. Th17 cells produce IL-17A, IL-17F and IL-22, which in turn stimulate macrophages and fibroblasts to produce more proinflammatory factors (IL-1, IL-6, TNF-α and PGE2) that trigger synovial inflammation [88].

Bone and cartilage destruction are induced by IL-17A in RA [63]. IL-17A also causes a reduction in bone formation and an increase in bone erosion by triggering the differentiation of osteoclast progenitors into mature osteoclasts and by promoting RANKL production by osteoblasts and synoviocytes [89,90,91]. IL-17A also causes cartilage destruction by stimulating synoviocytes to produce MMP-1 [92]. TNF-α, a product of synovial macrophages, B-cells and natural killer (NK) cells, plays an important role in mediating joint inflammation, bone resorption and cartilage destruction in RA [93,94,95]. TNF-α stimulates the production of inflammatory cytokines such as IL-1β and IL-6 by leukocytes and enhances synovial inflammation [93]. TNF-α can directly stimulate osteoclastogenesis or indirectly by stimulating RANKL secretion by osteocytes [96,97,98]. RANKL plays a significant role in bone regeneration and remodeling, as it binds to RANK to induce osteoclastogenesis [99]. In RA, bone destruction results from abnormal activation of osteoclasts induced by RANKL mainly produced by immune cells such as Th17 cells, macrophages, dendritic cells, activated B cells and FLS [100]. This is unlike normal physiological conditions, whereby RANKL is mainly produced by osteoblasts [101].

Tocotrienol has exhibited protective effects in countering the molecular mechanisms that induce the pathological changes seen in RA. Tocotrienol was reported to significantly reduce IL-17-activated expression of RANKL [55], RANKL-induced osteoclastogenesis [55] and TNF-α upregulation [52,54,55]. Tocotrienol also significantly suppressed the differentiation of osteoclasts in animal models of RA [55]. The proportion of IL-17+/CD4+ T cells in Th17 polarising conditions, wherein both Th17 cells and regulatory T cells were differentiated, was also reduced by tocotrienol [55]. Plasma levels of IL-1β and IL-6 were also significantly reduced upon tocotrienol treatment [52]. Lastly, tocotrienol also significantly reduced the levels of plasma C-reactive protein (CRP), which is an inflammatory marker [52,53,54].

A summary of the effects of tocotrienol on RA is presented in Figure 3B.

### 4.3. Prospectives and Research Gaps

Apart from being a standalone therapy, tocotrienol could complement current pharmacological agents and rehabilitation treatments that have already proven effective in patients with arthritis at the early stage [22,102,103]. However, the number of studies in this regard is rather limited. Only one study explored the effectiveness of combined palm tocotrienol and glucosamine sulphate regime in a rat model of osteoarthritis. The combination significantly improved the grip strength of the rats compared to tocotrienol alone. Similar effects were not observed in the circulating COMP level, perhaps because the maximal degree of suppression had been achieved [47]. The study is not conclusive, as the underlying mechanism of action remains elusive. Interestingly, tocotrienols were previously reported to possess antinociceptive and analgesic properties in both animal and clinical trials. Tocotrienol significantly reduced alcoholic neuropathic pain [104] and diabetic neuropathic pain (thermal hyperalgesia, mechanical hyperalgesia and tactile allodynia) [105]. Furthermore, tocotrienol reduced the lancinating pain score significantly among patients with diabetic neuropathy [106]. A dietary supplement blend with black tea extract, curcumin, resveratrol and tocotrienol reduced the pain among patients with chronic pain, although the individual effects of tocotrienol could not be determined [107]. Since pro-inflammatory cytokines are critical in pain sensitization [108], NSAIDs and paracetamol are used to control joint pain in patients with arthritis [109]. Chronic use of these medications could lead to various side effects. Tocotrienol has been shown to possess analgesic effects, probably through its anti-inflammatory activities. It may reduce the frequency of NSAIDs or paracetamol use pro re nata.

Despite the protective effects of tocotrienol on OA and RA, its application as an alternative supplement is hampered by its low bioavailability due to the hepatic presence of α-tocopherol transfer proteins, which favours the binding and transport into the circulatory system of α-tocopherol over tocotrienol [110,111]. Furthermore, delivering tocotrienol directly into the aqueous joint space is challenging, as tocotrienol is hydrophobic and the cartilage is avascular [9]. A more efficient method of administering tocotrienol to the joint space, such as interarticular injection together with a vehicle such as hyaluronic acid, must be studied. More studies on the bioavailability of vitamin E in the joint space are also warranted [112].

It is also important to note that the majority of the current studies on the protective effects of tocotrienol against the pathological changes observed in OA and RA focused on a post-treatment model instead of a pre-treatment model. In the cellular study by Pang et al., pre-treatment with tocotrienol before monosodium iodoacetate exposure did not prevent chondrocyte damage [49]. On the other hand, Kim et al. demonstrated that pre-incubation of PBMCs with tocotrienol significantly reduced RANKL-induced osteoclastogenesis [55]. In this context, the effect of tocotrienol when utilized in a pre-treatment model must be further investigated for the potential application of tocotrienol as a form of prophylaxis for arthritis, especially for at-risk individuals.

Postmenopausal women were disproportionately affected by OA [113], highlighting the important role of oestrogen deficiency in the development of OA [114]. However, both the in vivo studies of OA included in this review used male rats, and the effects of tocotrienol in castrated female models have not been investigated. This aspect should be investigated in the future to ensure tocotrienol is also effective at managing OA in postmenopausal women. The monosodium iodoacetate model is the only osteoarthritic model that has been used so far to test the joint protective effects of tocotrienol [47,48]. In comparison to surgical models such as meniscal transection, the monosodium iodoacetate model causes acute inflammation, osteophyte formation, osteochondral disruption and weight-bearing asymmetry [115]. Hence, the effects of tocotrienol on more severe osteoarthritic changes should be evaluated using appropriate osteoarthritic animal models.

We did not find any clinical trial related to tocotrienol treatment on RA. A total of six studies were found on ClinicalTrials.gov by using the keywords “tocotrienol” and “RA” [116]. Five of them are not related to tocotrienol, and the sixth (NCT00399282) is related to the omega-3 and vitamin E supplementation in patients with RA [117]. It is a phase 1 trial on 75 participants and was completed in April 2007. Nevertheless, in this study, tocotrienol was combined with omega 3, and its sole effects could not be determined. Additionally, only some of the findings (55 participants) were published, and it was written in a non-English language [118]. To the best of our knowledge, the full result has not been published.

### 4.4. Limitations of the Review

In this review, we did not evaluate the literature beyond the three electronic databases. We also did not search for grey literature or unpublished materials. Therefore, potential studies, especially negative results, could be missed. The reference list of the included articles was screened to minimize the missing literature. The number of studies uncovered from the search is limited, despite the use of a board search string, indicating that more intensive research on this topic would be necessary.

## 5. Conclusions

Preclinical studies showed that tocotrienol serves as a potential anti-arthritic drug, whereby it prevents structural changes in the joints and improves joint functions in animal models of OA and RA. It may also prevent osteolytic lesions of bones commonly found in early OA and RA. Tocotrienol also significantly reduced the inflammatory processes in RA models. Due to limited data from clinical trials, it cannot be confirmed that the anti-arthritic properties of tocotrienol will translate to patients with arthritis. More intensive research in this regard will be necessary to fill the void of suitable pharmacological agents in the market.

## Figures and Tables

**Figure 1 pharmaceuticals-16-00385-f001:**
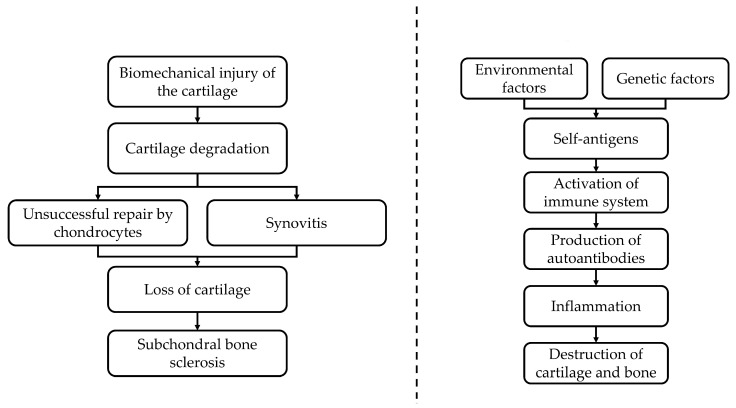
The pathogenesis of OA (**A**) and RA (**B**). OA is triggered by biomechanical injury and uncompensated healing of the cartilage layer, leading to local recurrent inflammation and the progressive destruction of the cartilage, and subsequently, subchondral bone remodeling. Rheumatoid arthritis is caused by systemic inflammation triggered by the production of autoantibodies, leading to the destruction of cartilage and bone. (**A**) Pathogenesis of Osteoarthritis. (**B**) Pathogenesis of Rheumatoid arthritis.

**Figure 2 pharmaceuticals-16-00385-f002:**
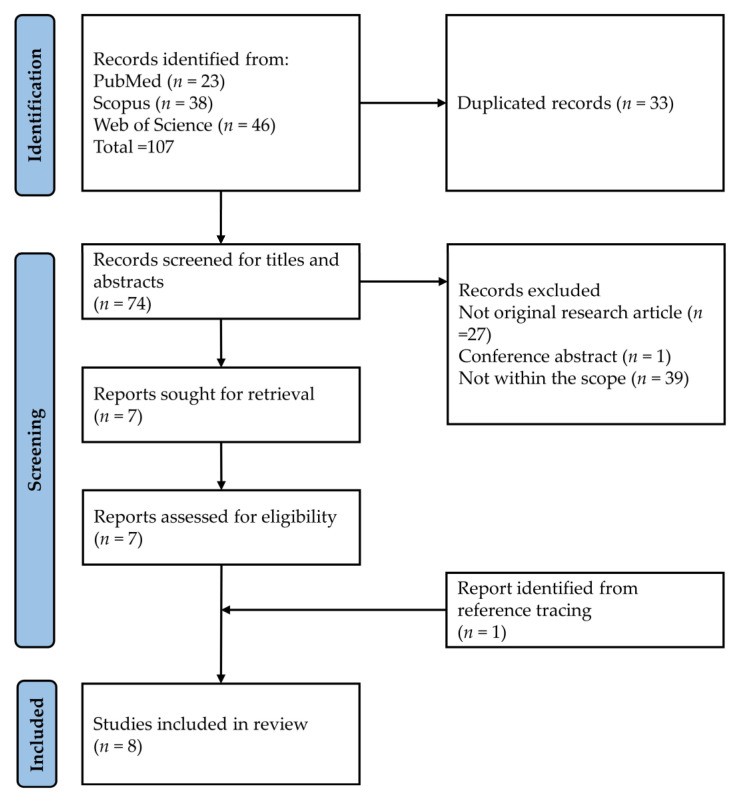
Article selection process.

**Figure 3 pharmaceuticals-16-00385-f003:**
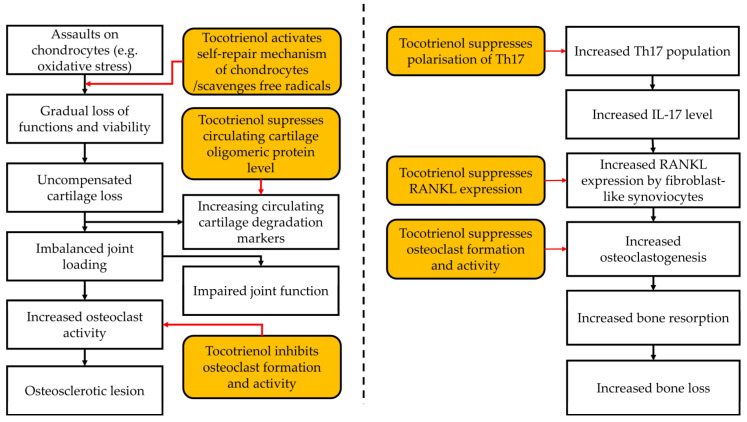
The effects of tocotrienol on OA and RA. (**A**) Effects of tocotrienol on osteoarthritis. (**B**) Effects of tocotrienol on rheumatoid arthritis.

**Table 1 pharmaceuticals-16-00385-t001:** The effects of tocotrienol on OA.

Researchers	Study Design	Findings
Pang et al. (2021) [49]	Disease model: SW1353 chondrocytes challenged with MIA Treatment: annatto tocotrienol (10–20 μg/mL) or palm tocotrienol (25–50 μg/mL) for 24 h before or concurrent with MIA	Cell viability:
		Pre-treatment
		Decreased upon receiving annatto tocotrienol (≥20 μg/mL) & palm tocotrienol (≥12.5 μg/mL) → no subsequent test
		Concurrent treatment
		Increased upon receiving annatto tocotrienol (10 and 20 μg/mL)/palm tocotrienol (3.125, 25 and 50 μg/mL)
		8-Isoprostane F2-α Level:
		Decreased upon receiving palm tocotrienol (50 μg/mL) or annatto tocotrienol (10 and 20 μg/mL) in the presence of MIA
		Collagen I type α1: Decreased in MIA group, not affected by treatment
		Collagen II type α1: Increased upon receiving annatto tocotrienol (10 & 20 μg/mL) in the presence of MIA vs MIA alone or annatto tocotrienol alone
		Collagen II type α1/Collagen I type α1: Increased upon receiving MIA + treatment vs treatment alone.
		Increased upon receiving annatto tocotrienol alone (10 and 20 μg/mL) vs. MIA alone
		Aggrecan and sex-determining region Y box protein 9:
		Increased upon receiving MIA + annatto tocotrienol (10 and 20 μg/mL) vs. MIA alone
		ADAMTS4: Decreased in MIA group, not affected by treatment
Chin et al. (2019) [48]	Animals: Male Sprague-Dawley rats (3 months old) Disease model: Intra-articular MIA injection at the right knee Treatment: annatto tocotrienol at the dose of 50, 100, 150 mg/kg/day, oral for 4 weeks Normal and OA control were given refined olive oil (vehicle in the study)	Body weight:
		Increased in all groups from week 1–week 4
		Histological scoring:
		Decreased in all aspects for 100 mg/kg/day annatto tocotrienol vs OA control
		Decreased in the number of inflammatory cells and synovial hyperplasia for 100 mg/kg/day vs 50 mg/kg/day annatto tocotrienol groups
		Reduced synovial hyperplasia and erosions in 150 mg/kg/day annatto tocotrienol group vs OA control
		No significant improvement in 50 mg/kg/day annatto tocotrienol group
		Serum COMP:
		Decreased in 50, 100 & 150 mg/kg/day annatto tocotrienol group vs OA control
		Decreased in 100 & 150 mg/kg/day T vs 50 mg/kg/day annatto tocotrienol
		Serum hyaluronic acid:
		Decreased in 50, 100 & 150 mg/kg/day annatto tocotrienol group vs OA control
		Subchondral osteoclast number:
		Decreased in 150 mg/kg/day annatto tocotrienol group vs OA control
		Serum osteocalcin:
		Increased in 50 mg/kg/day annatto tocotrienol group vs OA control.
		Decreased in 100 & 150 mg/kg/day annatto tocotrienol group vs 50 mg/kg/day annatto tocotrienol
		Decreased in 150 mg/kg/day annatto tocotrienol group vs all other groups except normal group
		Serum C-telopeptide of crosslinked collagen type I:
		No significant effect
Al-Saadi et al. (2021) [47]	Animals: Male Sprague-Dawley rats (3 months old) Disease model: Intra-articular MIA injection at the right knee Treatment: palm tocotrienol treated (100 mg/kg/day; oral) Glucosamine sulphate treated (250 mg/kg/day; oral) palm tocotrienol (100 mg/kg/day) + glucosamine sulphate-treated group (250 mg/kg/day) (oral) Normal and OA control were given refined olive oil (vehicle in the study) Treatment period; 4 weeks	Grip strength:
		No significant change between normal control, OA control & palm tocotrienol between week 0–week 4.
		Increased in palm tocotrienol + glucosamine sulphate group vs sham & palm tocotrienol group from week 1–week 3.
		Increased in glucosamine sulphate group in week 4 vs week 1.
		Increased in palm tocotrienol + glucosamine sulphate group in week 4 vs week 1–3.
		Body weight:
		Increased in normal control vs OA control, palm tocotrienol & palm tocotrienol + glucosamine sulphate groups at week 1.
		Increased in palm tocotrienol, glucosamine sulphate and combination group vs OA control at week 3.
		Cartilage histology (Mankin’s score):
		No significant change between OA control, palm tocotrienol, glucosamine sulphate and combination group.
		Serum COMP:
		Increased in OA control vs palm tocotrienol, glucosamine sulphate and combination group.
		No significant change between treated group.
Haflah et al. (2009) [46]	Subjects: 79 patients with knee OA (Kellgren-Lawrence score of 2 and 3) aged over 40 years Treatment: Oral palm tocotrienol (400 mg daily) for 6 months Positive control: Glucosamine sulphate (500 mg thrice daily) for 6 months Notes: Patients were not allowed to take any other analgesics	Visual analogue scale:
		Standing & walking
		Decreased in palm tocotrienol & glucosamine sulphate
		WOMAC score:
		Decreased in palm tocotrienol & glucosamine sulphate
		Serum malondialdehyde:
		Decreased in palm tocotrienol vs glucosamine sulphate
		Serum vitamin E:
		Increased in palm tocotrienol vs glucosamine sulphate

Abbreviations: ADAMTS4, a disintegrin and metalloproteinase with thrombospondin motifs 4; COMP, circulating cartilage oligomeric matrix protein; MIA, monosodium iodoacetate; WOMAC, Western Ontario and McMaster Universities’ OA index.

**Table 2 pharmaceuticals-16-00385-t002:** The effects of tocotrienol on RA.

Researchers	Study Design	Findings
Haleagrahara et al. (2014) [53]	Animals: Female Dark Agouti rats (6–10 weeks old) Disease model: Intradermal injection of a collagen-complete Freund’s adjuvant emulsion at the base of the tails of the rats (RA group) Treatment: δ-tocotrienol (oral, 10 mg/kg body weight) from day 25–50 post induction. Glucosamine hydrochloride (oral, 300 mg/kg body weight of treated) from day 25–50 post induction.	Mobility:
		Increased in glucosamine & δ-tocotrienol group vs RA group
		Paw oedema:
		Reduced in δ-tocotrienol & glucosamine group vs RA group
		Reduced in δ-tocotrienol group vs glucosamine group.
		Body weight:
		Increased in all groups from day 25–50.
		Decreased in δ-tocotrienol group vs RA group.
		Decreased in δ-tocotrienol & glucosamine group vs RA group after day 40.
		Histopathology:
		Decreased in the severity of arthritic joint changes in δ-tocotrienol & glucosamine group vs RA group.
		Reduced oedema, congestion & inflammation in δ-tocotrienol group vs RA group.
		Signs of healing present in δ-tocotrienol & glucosamine group.
		Reduced swelling in glucosamine group vs RA group.
		Reduced arthritic changes in glucosamine group. Signs of healing present in glucosamine group.
		Collagen induced proliferation of splenocytes:
		Reduced in δ-tocotrienol & glucosamine group vs RA group.
		Reduced in δ-tocotrienol group vs glucosamine group.
		CRP levels:
		Decreased in δ-tocotrienol & glucosamine group vs RA group.
		Decreased in δ-tocotrienol group vs glucosamine group.
Zainal et al. (2019) [52]	Animals: Female Dark Agouti rats (4–5 weeks old) Disease model: intradermal injection of collagen 2 emulsified in complete Freund’s adjuvant into each paw of the hind limbs (RA group). Two booster injections of the same concentration of ovalbumin were administered on day 7 and 14. Treatment: palm tocotrienol (oral, 30 mg/kg body weight) daily from day 21 until day 45 post-induction. The normal control and RA group were given refined bleached deodorised-stripped vitamin E oil, which was the vehicle in the study.	Final body weight:
		Increased in palm tocotrienol group vs RA group.
		Paw oedema:
		Decreased in palm tocotrienol group vs RA group.
		Mobility:
		Increased in palm tocotrienol group vs RA group.
		Joint histology:
		Reduced in the severity of arthritic changes in palm tocotrienol group vs RA group.
		Reduced cartilage erosion & degeneration in palm tocotrienol group vs RA group.
		Increased number of cartilage cells in palm tocotrienol group vs RA group.
		Reduced bone resorption in palm tocotrienol group vs RA group.
		Plasma levels of CRP, TNF-α, IL-1β, and IL-6:
		Reduced in palm tocotrienol group vs RA group.
		Bone destruction:
		Reduced in palm tocotrienol group vs RA group.
		Bone mineral density:
		Increased in palm tocotrienol group vs RA group.
Radhakrishnan et al. (2013) [54]	Animals: Female Dark Agouti rats (10 weeks old) Disease model: Intradermal injection of collagen-complete Freund’s adjuvant mixture into the 4 paws and tail of each rat (RA group) Treatment: γ-tocotrienol (oral, 5 mg/kg body weight) daily from day 21 to 45 post induction. It was not mentioned whether normal control or RA group were given vehicle.	Body weight:
		Increased in γ-tocotrienol group with time.
		Paw thickness:
		Decreased in γ-tocotrienol group vs RA group.
		Biochemical analysis:
		CRP: Decreased in γ-tocotrienol group vs RA group.
		TNF-α: Decreased in γ-tocotrienol group vs RA group.
		Glutathione: Increased in γ-tocotrienol group vs RA group.
		Superoxide dismutase: Increased in γ-tocotrienol group vs RA group.
		Histopathology:
		Reduced in grade of severity of arthritic changes in γ-tocotrienol group vs RA group.
		Reduced in joint space narrowing in γ-tocotrienol group vs RA group.
		Reduced in granulomatous accumulation in γ-tocotrienol group vs RA group.
		Reduced in inflammation in γ-tocotrienol group vs RA group.
Kim et al. (2021) [55]	Study 1: RA FLS from synovial tissues obtained from patients undergoing total knee replacement surgery and fulfilled the American College of Rheumatology classification of RA of the knee, with Kellgren-Lawrence grade 4 (as RA FLS) Study 2: Human PBMCs cultured for 48 h with anti-CD28 and anti-CD3, IL-23, IL-6, IL-1β, IL-4-blocking antibodies, and interferon-γ-blocking antibodies to induce Th17 differentiation. Then, to investigate the suppressive effects of tocotrienol, PBMCs were cultured for 3 h with tocotrienol and then incubated using the same method as Th17 differentiation. For RANKL signal pathway analysis, RA FLS were first cultured for 3 h with or without the addition of tocotrienol. Then, RA FLS were stimulated with IL-17 for 72 h.	The suppressive effect of tocotrienol on the IL-17 activated RANKL gene and protein in RA FLS:
		Reduced in IL-17-activated expression of RANKL.
		Reduced in TNF-α level
		IL-6 and IL-8 were unchanged in RA group vs treatment
		Signal pathways of tocotrienol:
		Reduced in mTOR, ERK, IκBα levels
		Increased of IL-17-activated phosphorylation of AMPK.
		Suppressive effect of tocotrienol in IL-17- and RANKL-activated osteoclast formation:
		Reduced in RANKL-induced osteoclastogenesis.
		Decreased in osteoclast markers (TRAP, cathepsin K, DC-STAMP, NF-ATc1 and OC-STAMP).
		The inhibitory effect of tocotrienol in osteoclast formation with coculture of monocytes in addition to RA FLS:
		Reduced in differentiation of osteoclasts.
		Downregulation in mRNA expression of osteoclast markers.
		The suppressive effects of tocotrienol on Th17 cell differentiation:
		Reduced in IL-17+/CD4+ T cell proportion.
		Not differentiated into CD25+Foxp3+/CD4+ regulatory T cells.
		Decreased in IL-17 and soluble RANKL levels

Abbreviations: AMPK, adenosine monophosphate protein kinase; CD, cluster of differentiation; CRP, C-reactive protein; DC-STAMP, dendritic cell-specific transmembrane protein; ERK, extracellular signal-regulated kinase; FLS, fibroblast-like synoviocytes; Foxp3, forkhead box P3; IL, interleukin; IκBα, inhibitor of kappa B-α; mTOR, mammalian target of rapamycin; NF-ATc1, nuclear factor of activated T cells 1; OC-STAMP, osteoclast stimulatory transmembrane protein; RANKL; receptor activator of nuclear factor kappa-Β ligand; Th17, T helper 17; TNF-α, tumor necrosis factor-α; TRAP, tartrate-resistant acid phosphatase.

## Data Availability

Not applicable.

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
