# Peer review of "The Role of Tocotrienol in Arthritis Management—A Scoping Review of Literature"

_pharmaceuticals, 2023, doi:10.3390/ph16030385_

Round 1

Reviewer 1 Report

Hi dear the editorial boards and the authors

This article " The use of tocotrienol in the management of arthritis – a scoping review of literature” was revised and has a novelty and I recommend it for publication after consideration of the following comments.

Title: If you can rewrite and make it more interesting for readers. It is better to use the word "improvement" etc., instead of the word "management".

·        Abstract: please address the functional role of tocotrienol in improving these diseases in a more specialized manner.

·        Abstract: be researched in abstract background about tocotrienol, rheumatoid arthritis, and osteoarthritis.

·        Line 116- 153: The explanations are very long and lack scientific validity in most cases. Please set it very briefly and of course scientifically.

·        I think it is better to put figure 1 in the methodology section.

·        There are many abbreviations in Tables  and it becomes difficult for the reader of the article to follow the content. It is better to be abbreviated increase or decrease and some words like T vs RA should be written in full.

·        Line 256-260 etc.: It is better to mention it in the introduction and there is no need to repeat it.

·        Please, for this article with this amount of abbreviations, the abstract section should be set, because the journal is against having it.

·        It seems that Tokotri Enol is not present only in (nnatto seed, palm kernel and rice bran) and it is better to state other sources more fully. I think you have dogmatically treated these sources only according to the results of your limited research. Please correct it.

·        Discussion text must grammar improve and in some cases it is very weak and maybe there is no discussion at all.

·        Conclusion is very general, try to make it more scientific, comprehensive and concise in detail, especially.

·        References: It is OK.

The article has many flaws in express and concept of English, it is suggested to be revised in a scientific and native way.

Author Response

Dear reviewer, 

Thank you for reviewing our manuscript. We appreciate the constructive comments given and have responded to each of them in the attached response sheet. We hope the edited manuscript can meet the standard of the esteemed journal. We look forward to receiving your favourable response.

Comment

Reply

This article " The use of tocotrienol in the management of arthritis – a scoping review of literature” was revised and has a novelty and I recommend it for publication after consideration of the following comments.

Thank you for the comments. We have revised the manuscript accordingly.

Title: If you can rewrite and make it more interesting for readers. It is better to use the word "improvement" etc., instead of the word "management".

Thank you for the comment. We honour the reviewer's opinion but wish to retain the word "management". This is because management carries a more specific meaning of prevention or treatment of the disease.

Abstract: please address the functional role of tocotrienol in improving these diseases in a more specialized manner.

Thank you for the comment. We limited the biological role of tocotrienol in the introduction of abstract to antioxidant and anti-inflammatory due to the word limits. However, the more specialized role of tocotrienol in RA and OA has been discussed in the results section of the abstract. We have mentioned that tocotrienol activates the self-repair mechanism of chondrocytes exposed to assaults and attenuates osteoclastogenesis associated with RA.

Abstract: be researched in abstract background about tocotrienol, rheumatoid arthritis, and osteoarthritis.

Thank you for the comment. We have provided a more detailed background for OA, RA and tocotrienol at the beginning of the abstract.

Line 116- 153: The explanations are very long and lack scientific validity in most cases. Please set it very briefly and of course scientifically.

Thank you for the comment. We humbly request the reviewer to reconsider this comment. The methodology section as pointed out by the reviewers is based on the scoping review framework of Arksey and O'Malley. It is written as such so that the literature search process is replicable. Similar examples of writing can be found in our published articles and others:

https://doi.org/10.1080/1364557032000119616

https://www.mdpi.com/2076-393X/10/6/930

https://www.mdpi.com/2072-6643/14/22/4851

https://www.eurekaselect.com/article/127935

I think it is better to put figure 1 in the methodology section.

Thank you for the comment. We have moved the PRISMA flow chart (Figure 2) to the methodology section.

There are many abbreviations in Tables and it becomes difficult for the reader of the article to follow the content. It is better to be abbreviated increase or decrease and some words like T vs RA should be written in full.

Thank you for the comment. We have reduced the number of unnecessary abbreviations in Table 1 and 2 as suggested.

Line 256-260 etc.: It is better to mention it in the introduction and there is no need to repeat it.

Thank you for the comment. We have deleted the paragraph.

Please, for this article with this amount of abbreviations, the abstract section should be set, because the journal is against having it.

We apologize for not being able to understand this comment. We have reduced some unnecessary abbreviations. 

It seems that Tocotrienol is not present only in annatto seed, palm kernel and rice bran and it is better to state other sources more fully. I think you have dogmatically treated these sources only according to the results of your limited research. Please correct it.

Thank you for the comment. We have expanded on the list of sources of tocotrienol as the following:

Line: Tocotrienol can be found naturally in varying compositions in oil derived from annatto seed, palm kernel, rice bran, sunflower seed, flax seed, poppy seed, grapefruit seed and others

Discussion text must grammar improve and in some cases it is very weak and maybe there is no discussion at all.

Thank you for the comment. We have improved and expanded the discussion. The discussion is constructed so that the effects of tocotrienol can be viewed in the context of pathogenesis of arthritis.

Conclusion is very general, try to make it more scientific, comprehensive and concise in detail, especially.

Thank you for the comment. We have amended the conclusion to make it sounds more scientific and comprehensive to readers

References: It is OK.

Thank you.

The article has many flaws in express and concept of English, it is suggested to be revised in a scientific and native way.

Thank you for the comment. We proofread the manuscript again to minimize grammatical or structural errors.

Thank you again for reviewing our manuscript carefully. We hope the edited manuscript can meet the standard of the esteemed journal. We look forward to receiving your favourable response.

Reviewer 2 Report

The present review article assessed the implications of tocotrienol in the management of OA and RA, by evaluating the literature in terms of cell culture models, animal models, and clinical studies. The topic is relevant and interesting, with a good contribution to a potential improvement in the management of these inflammatory diseases. However, a few changes are required in order to improve the present form of the paper. Specific requirements are listed below:

Abbreviations are explained when they first appear in the main text, even if they have been included in the abstract, and contribute to making the text easier to read and the information conveyed more efficiently. Once an abbreviation has been established and explained, it will be used throughout the entire manuscript, with the exception of the abstract, where it must be treated separately. Due to the fact that osteoarthritis and rheumatoid arthritis are used multiple times in the present manuscript, it is advisable to use their abbreviated forms (OA and RA). Please review the whole manuscript in terms of abbreviations.

L36-37 should not include the same bibliographic indexes in two consecutive sentences. Instead, it is recommended to search the literature and refer to new and relevant bibliographic references for each disease.

For better comprehension and readability, it is advisable to use a figure in which to include the most important aspects of the pathogenesis of OA and RA where tocotrienol may have implications. I suggest checking and referring to: PMID: 34831081

The current therapeutic management of OA and RA needs to be developed in the introduction section with the presentation of molecules to assess what the current options are and what shortcomings there are that could be addressed by using tocotrienol as adjuvant therapy. I suggest checking and referring to: PMID: 36058148 and PMID: 35386619.

The authors have described what they have done in the study, but they have not presented the novelty it brings to the field, or the precise reason for choosing this topic. Please revise this aspect in the last paragraph of the introduction.

L134 It is recommended to provide the name, version, country, city for all software used in the different analyses, while the link and date of access should be included as a bibliographic reference in the references section (EndnoteX9)

It is advisable, since reference is made to adjuvant therapies that can be used in arthritis, to briefly mention the possibility of combination with rehabilitative therapies. I suggest checking and referring to: PMID: 35454333.

It is advisable to introduce a PRISMA flow chart type instead of the design for figure 1 because it better synthesizes the information. A model can be found in Page, M.J.; et al. The PRISMA 2020 statement: An updated guide for reporting systematic reviews. Journal of Clinical Epidemiology 2021, 134, 178-189, doi:10.1016/j.jclinepi.2021.03.001.

Author Response

Dear reviewer,

Thank you for reviewing our manuscript. We appreciate the constructive comments given and have responded to each of them in the attached response sheet. We hope the edited manuscript can meet the standard of the esteemed journal. We look forward to receiving your favourable response. 

Dear reviewer,

Thank you for reviewing our manuscript. We appreciate the constructive comments given and have responded to each of them as the following:

Comment

Reply

The present review article assessed the implications of tocotrienol in the management of OA and RA, by evaluating the literature in terms of cell culture models, animal models, and clinical studies. The topic is relevant and interesting, with a good contribution to a potential improvement in the management of these inflammatory diseases. However, a few changes are required in order to improve the present form of the paper. Specific requirements are listed below:

 Thank you for the comments. We have considered the reviewer's comments and they are answered as the following.

Abbreviations are explained when they first appear in the main text, even if they have been included in the abstract, and contribute to making the text easier to read and the information conveyed more efficiently. Once an abbreviation has been established and explained, it will be used throughout the entire manuscript, with the exception of the abstract, where it must be treated separately. Due to the fact that osteoarthritis and rheumatoid arthritis are used multiple times in the present manuscript, it is advisable to use their abbreviated forms (OA and RA). Please review the whole manuscript in terms of abbreviations.

Thank you for the comments. We have replaced osteoarthritis and rheumatoid arthritis with the respective abbreviations (OA and RA) in the main text. We have proofread the manuscript and ensured consistency in the use of abbreviations.

L36-37 should not include the same bibliographic indexes in two consecutive sentences. Instead, it is recommended to search the literature and refer to new and relevant bibliographic references for each disease.

Thank you for the comment. We have cited different references for each sentence.

For better comprehension and readability, it is advisable to use a figure in which to include the most important aspects of the pathogenesis of OA and RA where tocotrienol may have implications. I suggest checking and referring to: PMID: 34831081

Thank you for the comment. We have included a basic schematic diagram of the pathogenesis of OA and RA in the introduction (Figure 1). We have further included Figure 2 to illustrate the mechanisms of action of tocotrienol on OA and RA in the discussion.

The current therapeutic management of OA and RA needs to be developed in the introduction section with the presentation of molecules to assess what the current options are and what shortcomings there are that could be addressed by using tocotrienol as adjuvant therapy. I suggest checking and referring to: PMID: 36058148 and PMID: 35386619.

Thank you for the comment. The current therapeutic management of OA and RA has been discussed in the manuscript (line 72-88). Besides, we emphasized the rationale for using tocotrienol as a disease-modifying antirheumatic drug or joint damage-preventing agent (line 89-93). In response to the reviewer’s comment, we further complemented the introduction section with the suggested articles. The possibilities of tocotrienol as a complementary therapy to support existing pharmacotherapy has been discussed in section 4.3

The authors have described what they have done in the study, but they have not presented the novelty it brings to the field, or the precise reason for choosing this topic. Please revise this aspect in the last paragraph of the introduction.

Thank you for the comment. We have restructured the last paragraph so that the rationale for reviewing this topic and the novelty of this review are highlighted.

L134 It is recommended to provide the name, version, country, city for all software used in the different analyses, while the link and date of access should be included as a bibliographic reference in the references section (EndnoteX9)

Thank you for the comment. We have supplemented the details of Endnote and provided the link to the software.

It is advisable, since reference is made to adjuvant therapies that can be used in arthritis, to briefly mention the possibility of combination with rehabilitative therapies. I suggest checking and referring to: PMID: 35454333.

Thank you for the comment. We have included a suggestion that tocotrienol could complement current rehabilitative therapies in section 4.3, but there is no study in this regard.

It is advisable to introduce a PRISMA flow chart type instead of the design for figure 1 because it better synthesizes the information. A model can be found in Page, M.J.; et al. The PRISMA 2020 statement: An updated guide for reporting systematic reviews. Journal of Clinical Epidemiology 2021, 134, 178-189, doi:10.1016/j.jclinepi.2021.03.001.

Thank you for the comment. The PRISMA flow chart used in the current review is based on the template released on PRISMA website.

Thank you again for reviewing our manuscript carefully. We hope the edited manuscript can meet the standard of the esteemed journal. We look forward to receiving your favourable response.

Round 2

Reviewer 2 Report

The authors improved their manuscript.